# Impact of Calcium and Potassium Currents on Spiral Wave Dynamics in the LR1 Model

**DOI:** 10.3390/e27070690

**Published:** 2025-06-27

**Authors:** Xiaoping Yuan, Qianqian Zheng

**Affiliations:** 1Information Engineering College, Hangzhou Dianzi University, Hangzhou 311305, China; xpyuan@hdu.edu.cn; 2Henan Joint International Research Laboratory of High Performance Computation for Complex Systems, School of Science, Xuchang University, Xuchang 461000, China

**Keywords:** spiral wave, arrhythmias, cardiac memory, action potential

## Abstract

Spiral wave dynamics in cardiac tissue are critically implicated in the pathogenesis of arrhythmias. This study investigates the effects of modulating calcium and potassium currents on spiral wave stability in a two-dimensional cardiac model. The gate variable that dynamically regulates the opening probability of ion channels also plays a significant role in the control of the spiral wave dynamics. We demonstrate that reducing gate variables accelerates wave propagation, thins spiral arms, and shortens action potential duration, ultimately inducing dynamic instability. Irregular electrocardiogram (ECG) patterns and altered action potential morphology further suggest an enhanced arrhythmogenic potential. These findings elucidate the ionic mechanisms underlying spiral wave breakup, providing both theoretical insights and practical implications for the development of targeted arrhythmia treatments.

## 1. Introduction

Spiral waves have been extensively identified as a fundamental mechanism driving lethal ventricular arrhythmias, as evidenced by experimental studies in animal hearts and clinical recordings in humans [1,2,3,4]. When spiral wave reentry occurs in the atria or ventricles, it may be implicated in arrhythmias such as atrial flutter or ventricular tachycardia [5,6,7,8]. The stability, wandering, and fragmentation of spiral waves in myocardial tissue can be modulated by altering the conductance of calcium and potassium channels [9,10,11,12]. Several mechanisms of spiral wave breakup have been proposed, including kinetic instability [13,14], heterogeneity in repolarization-induced fibrous block [15,16,17], and reduced tissue excitability [18].

The investigation of cardiac action potential (AP) dynamics through computational modeling has made significant strides in recent decades, particularly in unraveling arrhythmogenic mechanisms [19,20]. Among these, the phenomenon of cardiac memory, first described by Rosenbaum et al. in 1982 [21], has attracted considerable attention for its role in modulating arrhythmia susceptibility. Cardiac memory influences currents and ion channels, thereby affecting myocardial repolarization and refractory periods. Evidence suggests that this phenomenon is linked to ion channel remodeling [22,23], gap junction restructuring [24,25], and delayed membrane voltage coupling [26]. One such diseased condition is the presence of a strong transient outward potassium current ($I_{to}$), which can lead to a sudden shortening of action potential duration (APD), known as spike-and-dome AP morphology [27,28]. More recent work by Qu Zhilin et al. [29,30,31] has further associated cardiac memory with slow ion channel recovery kinetics and intracellular ion concentration accumulation, particularly in pathological conditions that promote dynamic instability.

Given the intrinsic complexity of cardiac tissue, the limitations of existing control strategies, and the incomplete understanding of how cardiac memory influences excitation wave dynamics in heterogeneous substrates, this study investigates spiral wave stability in a cardiac AP model incorporating multi-timescale ion channels. Our objective is to elucidate the mechanisms underlying spiral wave breakup. We demonstrate that accelerating the dynamics of specific variables can stabilize spiral waves under certain conditions, whereas increasing the maximum conductance of key channels promotes breakup. Furthermore, we show that APD restitution properties can be modulated through conductance adjustments, offering new avenues for arrhythmia management.

## 2. Materials and Methods

We carry out single-cell and two-dimensional (2D) tissue simulation with the 1991 Luo–Rudy (LR1) porcine ventricular AP model [32], which is a continuous approximation to cardiac tissue. The following partial differential equation is for the single-cell model:(1)$$\frac{dV}{dt}=\left(-I_{ion}+I_{sti}\right)/C_{m}$$
where V is the transmembrane potential, $C_{m}=1\mu F/{cm}^{2}$ is the membrane capacitance. $I_{ion}$ is the total ionic current density, and $I_{sti}$ is the external stimulation current density. $I_{ion}=I_{Na}+I_{Ca,L}{+I}_{K}{+I}_{K1}{+I}_{Kp}{+I}_{b}{+I}_{to,f}$, in which $I_{to,f}=G_{to.f}\cdot x_{to,f}\cdot y_{to,f}\cdot (V-E_{K})$ is the fast component of the transient outward potassium current ${(I}_{to})$ taken from the model by Mahajan et al. [33]. In addition, the other ion current and gate variables have the same form as the Hodgkin–Huxley model [34,35]. The form of the time evolution equation for the ion current is $I_{\beta }=G_{\beta }\cdot y\cdot (V-E_{\beta })$, where $G_{\beta }$ is the maximum conductivity of the ion current and the $E_{\beta }$ is the energy potential of the corresponding ion, that is, the potential difference between inside and outside the cell. Here, y is the function of the corresponding ion gate variable (y = m, h, j, d, f, x), which satisfies the differential equation $dy/dt=(y_{\infty }-y)/\tau_{y}$. The corresponding time constants are denoted by $\tau_{y}$, which is used to regulate the speed of opening and closing of ion channels.

The differential equation for voltage in 2D tissue is as follows:(2)$$\frac{\partial V}{\partial t}=\frac{-I_{ion}}{C_{m}}+D(\frac{\partial^{2}V}{\partial x^{2}}+\frac{\partial^{2}V}{\partial y^{2}})$$
where D is the diffusion constant. We used $D=0.001\;{cm}^{2}/ms$ as the original value, which makes the conduction velocity of the LR1 model about 0.55 m/s and the maximum $I_{Na}$ conductivity of $16\;mS/{cm}^{2}$. We use no-flux boundary conditions.

We used a typical cross-field protocol to induce individual 2D spiral waves. The initial conditions were set as follows. Single cells (Equation (1)) are stimulated 1000 times at 500 ms for a pacing cycle (PCL) to eliminate transient behaviors. The values of all variables recorded at the end of pacing were used as the initial conditions for all cells in 2D tissues. APD is calculated from cells located in the middle. APD is defined as the period of time during which voltage V remains above −75 mV. The tissue is discretized to 1024 × 1024 cells with ∆x=∆y=0.0125 cm, and the time step is ∆t = 0.01 ms.

## 3. Results

Ionic current and its dynamics have been widely demonstrated to play a key role in spiral wave stability, when the $I_{to}$ is absent, the spiral wave remains stable for a wide range of parameters. Based on patch-clamp experimental data, Luo and Rudy recalculated L-type calcium ion currents and developed the LRd model in 1994 to show data for faster kinetics [36]. However, the kinetics of this formulation are slow; thus, the current was called the slow-inward current ($I_{Si}$). As shown in previous work [37], faster kinetics can stabilize the spiral wave by restoring the APD relaxation. In addition to the maximum conductance of the calcium ion current $G_{Si}$, the corresponding time constant of activation and inactivation, denoted by $\tau_{d}\;\mathrm{and}\;\tau_{f}$, will accelerate its ion current and is an important parameter for controlling the dynamics of the spiral wave.

To gain further insights into the gate variables promoting spiral wave stability in the LR1 model, we carried out simulations to investigate the effects of the calcium ion current conductance $G_{si}$ and the gate variables $\tau_{d}\;\mathrm{and}\;\tau_{f}$ on spiral dynamics. Figure 1 shows a phase diagram of $G_{si}$ and $\tau_{d},\tau_{f}$ in the case of accelerated dynamics by reducing the gate variable values to below 50%. In the absence of $I_{to}$, it is found that $G_{si}=0$ or very small, the spiral wave was almost stable or showed quasi-periodic, as seen in the left region of Figure 1A. When $G_{si}$ exceeds a certain threshold, the system will change to a space–time chaotic state from a stable spiral wave. When $\tau_{d}\;\mathrm{and}\;\tau_{f}$ are very small, the system is in a state of stable spiral waves, except when $G_{si}$ is larger. Spiral wave breakup occurs when $G_{si}\geq 0.09\;mS/{cm}^{2}$ at $\tau_{d},\tau_{f}=0.1$ and $G_{si}\geq 0.11\;mS/{cm}^{2}$ at $\tau_{d},\tau_{f}=0.2$ (as shown in the lower region). The increase in calcium ion current stimulates more excitable cells and makes the spiral wave unstable. However, the larger $G_{si}$ can prevent the breakup of spiral wave when the time constants of activation and inactivation reduce by 70% or 60% of the control value, i.e., $\tau_{d},\tau_{f}=0.3\;\mathrm{or}\;0.4$. We can see that the system changes from a space–time chaotic state to a stable spiral wave as $G_{si}$ exceeds the threshold. The representative voltage snapshots from these three regions are shown in Figure 1B–D, which are marked by arrows in Figure 1A. When $G_{si}$ is small, a spiral wave in 2D tissue is stable with shortwave lengths, as smaller calcium ion current leads to a smaller repolarization region. With the increase of $G_{si}$, spiral wave breakup occurs. When $G_{si}$ is larger, the system returns to a stable, thicker spiral wave state with longer wavelengths.

Now we incorporate an $I_{to}$ current into the model, which is based on the formulation of the fast $I_{to,f}$ from Mahajan et al.’s model. It has been widely shown that $I_{to}$ and its kinetics play a crucial role in spiral wave stability [38,39]. However, the underlying mechanism remains poorly understood. To gain deeper insights into how this current promotes spiral wave breakup in the LR1 model, we systematically investigated the interactions between this current and other parameters. In the case of the original kinetic formulation, $\tau_{d},\tau_{f}=1$, as shown by the red arrow in Figure 2A, spiral wave breakup can be prevented by either reducing the parameter $G_{si}$ or increasing $G_{to}$. Moreover, the critical $G_{to}$ required to prevent its breakup monotonically increases with an increase in $G_{si}$. The asterisk-marked position indicates the original parameter control value of the LR1 model.

In the scenario of speedup kinetics, two dynamics were simulated with an 80% and 70% reduction in both the activation and inactivation time constants. In the $\tau_{d},\tau_{f}=0.2$ case, regardless of the presence of $I_{to}$ current, the system keeps a stable spiral wave until $G_{si}$ becomes very large. Only when $G_{si}\geq 0.1$ does the $I_{to}$ current have any effect on spiral dynamics. $I_{to}$ current can promote spiral wave breakup when $G_{si}=0.1$, as we can see in Figure 2B, and then transition to stability again as $G_{to}$ increase. This phenomenon is more interesting in the following cases $\tau_{d},\tau_{f}=0.3$ (Figure 2C). When $G_{si}$ is large and $G_{to}$ is small, spiral waves are stable, see the upper left area of Figure 2C. As $G_{to}$ increases, spiral waves break down. When $G_{to}$ is even larger, spiral wave stabilization reappears. The phase space of spiral wave breakup is compressed within a narrow parameter channel. As an example, voltage snapshots of spiral waves at different $G_{si}$ and $G_{to}$ levels are shown in Figure 3. Under the action of $I_{to}$, increasing the $I_{Si}$ current can cause it to break up (Figure 3A,C) or promote spiral wave stabilization (Figure 3B,D). These simulations provide a clearer understanding of the interplay between the currents and their effects on spiral wave behavior.

The detailed dynamic evolution of spiral waves in the case of speedup dynamics is shown in Figure 4. First of all, the entire left boundary of cells in the tissue was paced, triggering a wave to propagate towards the center of the tissue. Upon the cell in the middle of the tissue first depolarized above −55 mV and then dropped below −55 mV, the computer program then forces all cells in the bottom half of the tissue to attain a voltage of exactly −30 mV for 2 ms. It successfully produced spiral waves in the middle of the tissue.

When there is no $I_{to}$, the system can maintain spiral waves. We find that the spiral wave has a thick arm; that is, the voltage repolarization region of 2D tissue is wide, as seen in Figure 4A. When increase $G_{to}$ above a critical value—$G_{to}=0.11\;mS/\mu F$—spiral wave breakup occurs. After a period of evolution, the center of the spiral wave began to break into multiple tip points. They collide with each other and gradually cause the entire system to enter a chaotic state (Figure 4B). Continue to increase $G_{to}$ above the next critical value—$G_{to}=0.44\;mS/\mu F$—the system will transition to a stable spiral wave again (Figure 4C). At this time, the action voltage of two-dimensional tissue repolarization decreases significantly (red represents the highest voltage), and the spiral wave arm becomes thinner and denser. The propagation speed of the spiral wave increases significantly, as we can see that the depolarization wave has traveled to the center of the medium at t = 160 ms (Figure 4C); however, it is 330 ms without $I_{to}$ current (Figure 4A).

To gain more mechanistic insights into the effects of $I_{to}$ on promoting spiral wave breakup, we simulated pseudo-ECGs from different viewing angles when myocardial tissue is inserted into single and double electrodes. Figure 5 presents pseudo-ECGs at different $G_{to}$, with calcium ion conductance illustrated separately in the following images: $G_{si}=0.08\;mS/\mu F$ (seen in Figure 5A,B) and $G_{si}=0.09\;ms/\mu F$ (seen in Figure 5C,D). As $G_{to}$ increases to a certain threshold, the resulting pseudo-ECG becomes highly irregular, mainly due to $I_{to}$ promoting the spatiotemporal irregularity in spiral wave dynamics. When $G_{to}$ is further increased to 0.28 mS/μF in Figure 5B, the ECG stabilizes again, which is consistent with the result of spiral wave stabilization again at this point. However, it is important to note from Figure 5C that the ECGs remain highly irregular even at $G_{to}=0.28\;mS/\mu F$. The critical value of $G_{to}$ for the phase diagram transition exhibits a direct proportionality with the enhancement of $G_{si}$. This indicates a significant relationship between these two parameters within the scope of our study, suggesting that unstable spatiotemporal spiral mechanics may indeed be a lethal cause of ventricular arrhythmias.

AP dynamics simulations of single cells were also carried out to gain more mechanistic insights into the effect of spiral wave stability. In the myocardial model, the slope of the APD restitution curve is an important parameter that determines the spiral wave stability, which has a sensitive dependence on certain ion currents. For the control kinetics (Figure 6A), the APD restitution curve transitions from a monotonic function to a non-monotonic one as $G_{to}$ increase. It increases to approximately 450 ms and subsequently decreases rapidly to 350 ms, where a plateau phase becomes apparent Its shifting to the right reveals that $I_{to}$ uncovers or exacerbates the effect of memory, resulting in APD strongly depending on the previous pacing history. For the speedup kinetics as seen in Figure 6B, the APD restitution curve is flat, and the action potential is reduced significantly compared to the control case at $G_{to}=0$. Increasing $G_{to}$ initially decreases APD slightly (dashed red curve in Figure 6B), but then it decreases sharply from 200 ms down to 50 ms when $G_{to}$ reaches a certain value (dotted blue trace in Figure 6B). The propagation speed of the spiral wave increases significantly, as we can see in Figure 4C, which results in short APD.

The presence of $I_{to}$ causes significant changes in AP morphology. The notch of the first stage of AP is aborted at about 10 mV under the original control parameters ($G_{to}=0$, in Figure 7A), while $I_{to}$ creates a more obvious notch to −30 mV at $G_{to}=0.24\;mS/\mu F$ in Figure 7B. The voltage drops down to the notch and then bounces back before depolarization, known as a spike and dome AP pattern. When $G_{to}$ is greater than the critical value ($G_{to}=0.278\;mS/\mu F$), the voltage drops and repolarizes immediately without the normal second-stage platform, resulting in early repolarization. The obvious feature of early repolarization is the significant shortening of APD. In particular, the cellular APD alternans can be caused by the presence of certain $I_{to}$ (Figure 7D), which may account for T-wave alternans in patients with Brugada syndrome [40,41]. What effect would an increase in PCL have on APD alternans? As seen in Figure 7E, APD alternans can be eliminated during a slow pacing period of 1200 ms. This mechanism of APD alternans is due to steep APD restitution resulting from the recovery of the ionic current.

## 4. Discussion

In summary, we investigated the mechanism of spiral wave breakup and the effects of $I_{to}$ on spiral wave dynamics in a 2D cardiac tissue. Spiral wave stability was studied by altering the maximum conductance of Ca^2+^ and outward K^+^ currents in the LR1 model, particularly in cases of accelerated dynamics by reducing the gate variables $\tau_{d},\tau_{f}$. The results indicate that the gate variable plays a significant role in controlling spiral wave dynamics. We discuss the mechanisms by which spiral waves rupture and $I_{to}$ promote spiral wave splitting. As $G_{to}$ increases, the action voltage of two-dimensional tissue repolarization decreases significantly, and the spiral wave arm becomes thinner and denser with increased propagation speed. The irregular changes in the ECG in various directions and AP morphology also suggest that $I_{to}$ promotes dynamic instability. The significant changes in AP morphology further confirm the influence of $I_{to}$ and PCL. We hope this may help people decide whether it is necessary to intervene—or even prevent—the phenomenon of cardiac memory, and provide a certain basis for clinical treatment.

## Figures and Tables

**Figure 1 entropy-27-00690-f001:**
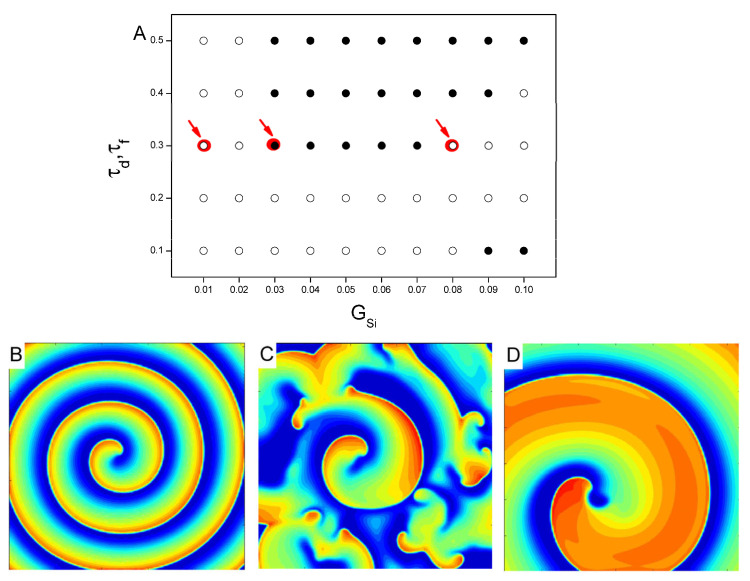
(**A**) Dynamical behavior of the phase space system at the maximum conductance G_si_ and time constant $\tau_{d},\tau_{f}$. Open circles are marked as spiral waves without breakup, and solid circles are marked as breakup. In the case of the corresponding kinetic state with $\tau_{d},\tau_{f}=0.3$, PCL = 2000 ms; (**B**) $G_{si}=0.01\;mS/\mu F$; (**C**) $G_{si}=0.03\;mS/\mu F$; (**D**) $G_{si}=0.08\;mS/\mu F$.

**Figure 2 entropy-27-00690-f002:**
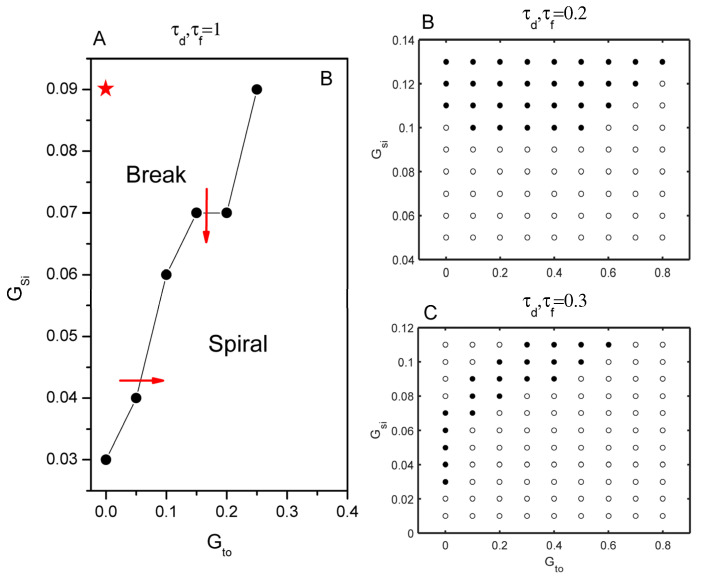
(**A**) The spiral wave behavior in the parameter space of $G_{si}$ and $G_{to}$ in the original $I_{si}$ kinetic system, and the star represents the original control parameter values. (**B**) The spiral wave behavior in the parameter space of $G_{si}$ and $G_{to}$ in the accelerated $I_{si}$ kinetic system, with a time constant $\tau_{d},\tau_{f}=0.2$. (**C**) Time constant $\tau_{d},\tau_{f}=0.3$.

**Figure 3 entropy-27-00690-f003:**
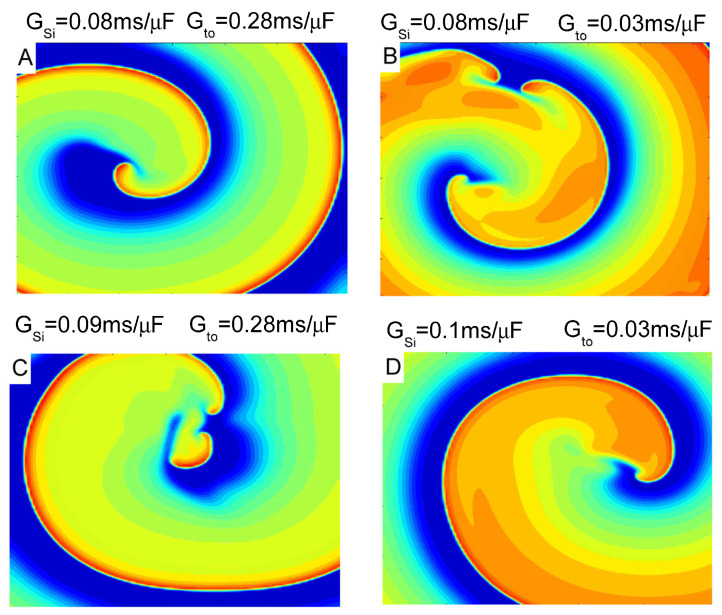
Spiral wave voltage snapshots at different parameters in the case of acceleration dynamics. Time constant $\tau_{d},\tau_{f}=0.3$. (**A**) $G_{si}=0.08\;mS/\mu F,G_{\mathrm{to}}=0.28\;mS/\mu F;$ (**B**) $G_{si}=0.08\;mS/\mu F,G_{to}=0.03\;mS/\mu F;$ (**C**) $G_{si}=0.09\;mS/\mu F,G_{to}=0.28\;mS/\mu F;$ (**D**) $G_{si}=0.1\;mS/\mu F,G_{to}=0.03\;mS/\mu F$.

**Figure 4 entropy-27-00690-f004:**
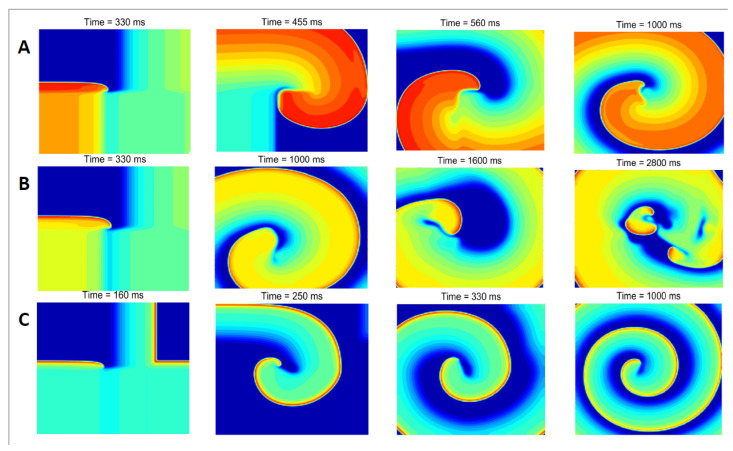
Dynamical evolution of spiral waves over time in accelerating dynamics at different $G_{to}$. The other parameters are $\tau_{d},\tau_{f}=0.3$, $G_{si}=0.09\;mS/\mu F$, and PCL = 2000 ms. (**A**) $G_{to}=0\;mS/\mu F;$ (**B**) $G_{to}=0.17\;mS/\mu F;$ (**C**) $G_{to}=0.51\;mS/\mu F$.

**Figure 5 entropy-27-00690-f005:**
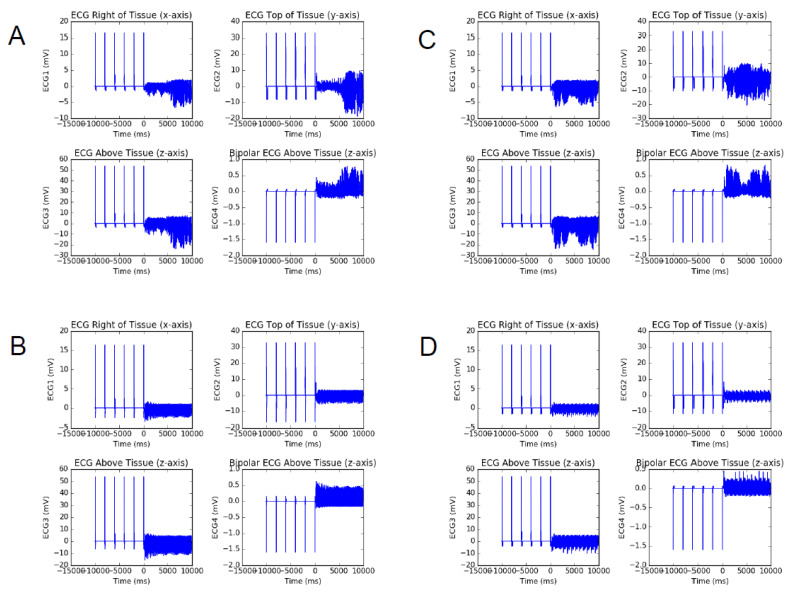
Pseudo-ECGs in different viewing angles. (**A**) $G_{si}=0.08\;mS/\mu F$, $G_{to}=0.17\;mS/\mu F$; (**B**) $G_{si}=0.08\;mS/\mu F$, $G_{to}=0.28\;mS/\mu F$; (**C**) $G_{si}=0.09\;mS/\mu F$, $G_{to}=0.28\;mS/\mu F$; (**D**) $G_{si}=0.09\;mS/\mu F$, $G_{to}=0.51\;mS/\mu F$. ECG1: a monopolar pseudo-ECG reading from the right edge of the tissue; ECG2: a monopolar pseudo-ECG reading from the top edge of the tissue; ECG3: a monopolar pseudo-ECG reading from above the tissue; ECG4: a bipolar pseudo-ECG reading from above the tissue. Two probes are placed very close to each other above the tissue.

**Figure 6 entropy-27-00690-f006:**
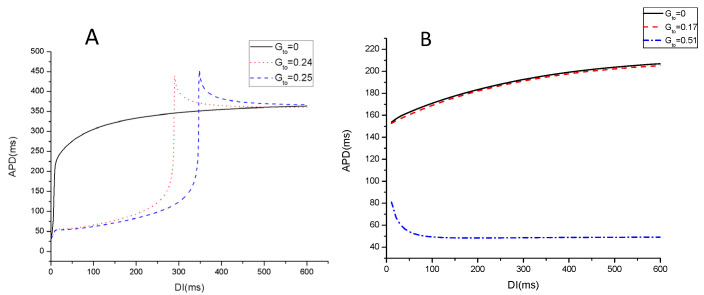
Restitution curves showing the dependence of APD on the DI at different $G_{to}$. (**A**) $\tau_{d},\tau_{f}=1$, $G_{si}=0.09\;mS/\mu F$; (**B**) $\tau_{d},\tau_{f}=0.3$, $G_{si}=0.09\;mS/\mu F$.

**Figure 7 entropy-27-00690-f007:**
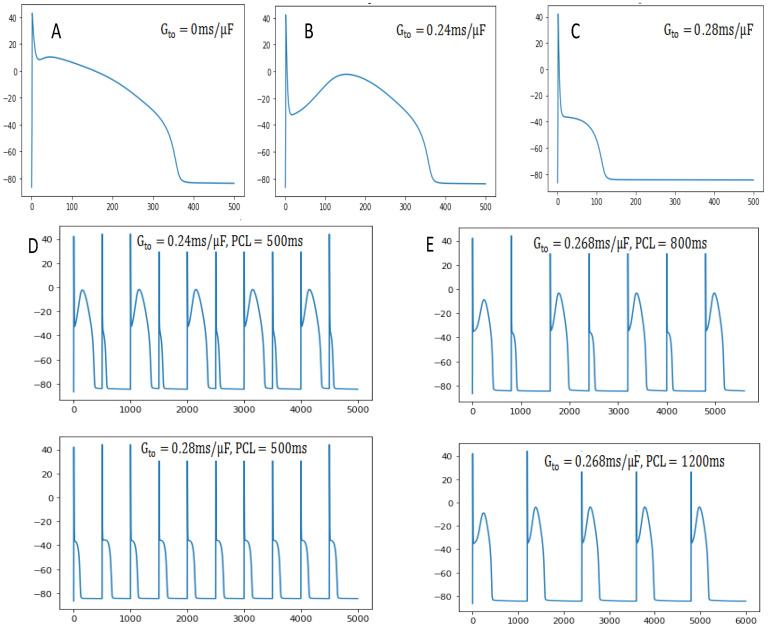
Action potential AP curves at different $G_{to}$ and PCL. (**A**) $G_{to}=0\;mS/\mu F$; (**B**) $G_{to}=0.24\;mS/\mu F$; (**C**) $G_{to}=0.28\;mS/\mu F$; (**D**) $G_{to}=0.24\;mS/\mu F$ and $G_{to}=0.28\;mS/\mu F$, PCL = 500 ms; (**E**) PCL = 600 ms and PCL = 1200 ms, $G_{to}=0.268\;mS/\mu F$.

## Data Availability

Data is contained within the article.

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
