# Peer review of "Impact of Calcium and Potassium Currents on Spiral Wave Dynamics in the LR1 Model"

_entropy, 2025, doi:10.3390/e27070690_

Round 1
Reviewer 1 Report
Comments and Suggestions for Authors
The mechanism of spiral wave instability in the LR1 model were investigated, which includes numerous ionic current equations and accurately simulates the complexity of myocardial tissue. The paper provides insights into how outward potassium ion currents and calcium ion currents affect spiral wave dynamics under different acceleration kinetics parameters. A possible explanation to these phenomena is also given. The conditions and the results are novel and interesting. I suggest this paper to be published after a minor revision.
- The unit of Gsi in Figure 1 is different from that used elsewhere in the entire document. Please verify and make the necessary adjustments.( line 212)
- The description of the initial conditions could be more organized. (lines 74-80) For example, ”APD is defined as the period of time when voltage V remains above minus 75mV”could be rewritten as ”APD is defined as the period of time when voltage V remains above -75mV”?
- The results section demonstrates the effects of different ionic currents and gating variables on spiral wave stability through abundant charts, with relatively detailed data. However, parameter annotations and coordinate descriptions in some figures (e.g., Figure 1, Figure 2) are unclear, potentially affecting readers' understanding of the results. It is necessary to optimize the readability of the figures by clarifying the meanings and units of key parameters.
- The overall structure of the manuscript is complete, but some formula typesetting issues exist (e.g., irregular formatting in Equations 1 and 2). Additionally, some English abbreviations (e.g., APD, ECG) lack clear Chinese full names when first introduced. These should be standardized according to academic paper formatting requirements to enhance the paper's rigor and readability.
Author Response
Comments 1:
The mechanism of spiral wave instability in the LR1 model were investigated, which includes numerous ionic current equations and accurately simulates the complexity of myocardial tissue. The paper provides insights into how outward potassium ion currents and calcium ion currents affect spiral wave dynamics under different acceleration kinetics parameters. A possible explanation to these phenomena is also given. The conditions and the results are novel and interesting. I suggest this paper to be published after a minor revision.
- The unit of Gsiin Figure 1 is different from that used elsewhere in the entire document. Please verify and make the necessary adjustments.( line 212)
- The description of the initial conditions could be more organized. (lines 74-80) For example, ”APD is defined as the period of time when voltage V remains above minus 75mV”could be rewritten as ”APD is defined as the period of time when voltage V remains above -75mV”?
- The results section demonstrates the effects of different ionic currents and gating variables on spiral wave stability through abundant charts, with relatively detailed data. However, parameter annotations and coordinate descriptions in some figures (e.g., Figure 1, Figure 2) are unclear, potentially affecting readers' understanding of the results. It is necessary to optimize the readability of the figures by clarifying the meanings and units of key parameters.
- The overall structure of the manuscript is complete, but some formula typesetting issues exist (e.g., irregular formatting in Equations 1 and 2). Additionally, some English abbreviations (e.g., APD, ECG) lack clear Chinese full names when first introduced. These should be standardized according to academic paper formatting requirements to enhance the paper's rigor and readability.
response 1:
We sincerely appreciate the time and effort you have invested in reviewing our work. Your insightful comments have significantly helped us improve the quality of our paper. Below, we provide detailed responses to each of your comments and concerns.
Our reply:
- 1) We sincerely apologize for this omission. As the reviewer correctly pointed out, the conductance values in the LR1 model are normalized by membrane capacitance (mS/μF). We have now clarified this in line 212. Some models, such as Hodgkin-Huxley, may directly use mS/cm².
- 2) It has been rewritten as “-75mV”. Thank you for pointing this out.
- 3) The description of the parameters has been added in the main body of the article. The definition of the time constant is stated in line 65 to line 72, and the supplementary explanations of ISi and are newly provided from line 95 to line 96.
- 4) Equation 1 and Equation 2 have already been standardized in format. The full name of ECG and APD can be seen in Line 18 and line 52.

Reviewer 2 Report
Comments and Suggestions for Authors
In this paper, the dynamics of spiral waves in 2D cardiac tissue has been investigated. They found that the wave propagation can be accelerated by reducing the gate variables. The phenomena can be interpretated by the pseudo-ECGs patterns and altered action potential morphology. Overall, the manuscript is well-prepared, and can be accepted after addressing the following points:
- The potential v should be capital V. What are the Na, K, K1, Kp, b currents, and how do they depend on m, n, h… The HH equations should be added in the manuscript. Also all the parameters should be defined.
- What is the \tau_d, \tau_f?
- To have a better presentation of the phase diagram Fig. 1A, it is better to define the two phases, what is bursting?
- All the plots should be in the same format, the same font, size, et. al.
Author Response
We sincerely appreciate the time and effort you have invested in reviewing our work. Your insightful comments have significantly helped us improve the quality of our paper. Below, we provide detailed responses to each of your comments and concerns.
Comments 1:The potential v should be capital V. What are the Na, K, K1, Kp, bcurrents, and how do they depend on m, n, h… The HH equations should be added in the manuscript. Also all the parameters should be defined.
Response 1: The description of the parameters has been added in the main body of the article, as seen in line 60-72. The definition of the time constant is stated in line 71 to line 72, and the supplementary explanations of ISi and are newly provided from line 95 to line 99.
Comments 2:What is the \tau_d, \tau_f?
Response 2 : GSi is the maximum conductance of the ion current of Isi, and \tau_d, \tau_f denote the corresponding time constant of activation and inactivation. For more detailed sources, please refer to the reference [20].
Comments 3:To have a better presentation of the phase diagram Fig. 1A, it is better to define the two phases, what is bursting?
Response 3 :The two phases in Fig.1 have been newly defined in line 98 to line 99. Bursting should be changed to breakup. Thank you for pointing this out.
Comments 4: All the plots should be in the same format, the same font, size, et. al.
Response 4: All the plots have been adjusted . The source files for all figures have been provided and further adjustments in accordance with the journal's requirements will be made with the help of the editor.

Reviewer 3 Report
Comments and Suggestions for Authors
The authors investigated the dynamics of spiral structures in a 2D system using the Luo-Rudy porcine ventricular action potential model. They introduced a new term I_{to,f} in the original model, which is suggested in the paper by Mahajan et al. The authors performed numerical simulations by changing the parameters and discussed the effect of I_{to} on the dynamic instability. The results seem important, but the manuscript is not clear mainly due to the lack of description of the I_{to}. Moreover, Entropy is not a specific journal on the behavior of the action potential. Therefore, I cannot recommend the present manuscript for publication in Entropy.
1) I have checked the paper by Mahajan and I found that the term I_{to,f} is suggested by Shannon et al. Only reading the present manuscript, I cannot catch the meaning of the new term I_{to,f}. The authors should clarify the meaning of the new term I_{to,f}.
2) In the introduction, the authors wrote "Our objective is to elucidate the mechanisms underlying spiral wave breakup and explore potential therapetutic targets."
3) The title of the manuscript includes "Ito-induced spiral wave breakup". Of course, the LR1 model is famous in the field of action potentials in cardiac tissues. However, it is not proper to only mention the "LR1 model" in Entropy. Moreover, many authors cannot understand the meaning of "Ito-induced". They should reconsider a proper title for Entropy.
4) The character v in Equation (1) should be capitalized.
5) The subsection "3.1) Subsection" is not necessary.
6) The authors should comment that they assume that tau_d = tau_f.
7) I cannot catch the meaning of the sentence "Ito can promote spiral wave breakup or stable in AP models." on the 120th line.
8) I do not know why the authors chose the parameters in Figure 3: G_si and G_to are both varied and thus the dependency is not clear. Moreover, is the pattern in Figure 3c stable? I found the small structure near the spiral core.
Author Response
Thank you for providing valuable feedback on our manuscript. We sincerely appreciate the time and effort you have invested in reviewing our work. Your insightful comments have significantly helped us improve the quality of our paper.Our paper is submitting to the special issue of Entropy named “Emergent Dynamics of Complex Systems”.The I_to has been described in line 63-65. It is hoped that readers in this field will be able to understand clearly.
Below, we provide detailed responses to each of your comments and concerns.
Comments 1: I have checked the paper by Mahajan and I found that the term I_{to,f} is suggested by Shannon et al. Only reading the present manuscript, I cannot catch the meaning of the new term I_{to,f}. The authors should clarify the meaning of the new term I_{to,f}.
Response 1: In the paper by Mahajan , they pointed out they used the original formulation by Luo and Rudy as subsequently implemented in the Shannon et al. In our work, we referred to the equations in the appendix of this paper for numerical simulation as shown in below. So relevant literature was cited to address the necessary corrections. (line 122)
Comments 2: In the introduction, the authors wrote "Our objective is to elucidate the mechanisms underlying spiral wave breakup and explore potential therapetutic targets."
Response 2:We have delete “explore potential therapetutic targets”. And It is hoped that it can be achieved in future work.
Comments 3: The title of the manuscript includes "Ito-induced spiral wave breakup". Of course, the LR1 model is famous in the field of action potentials in cardiac tissues. However, it is not proper to only mention the "LR1 model" in Entropy. Moreover, many authors cannot understand the meaning of "Ito-induced". They should reconsider a proper title for Entropy.
Response 3:Our paper is submitting to the special issue of Entropy named “Emergent Dynamics of Complex Systems”. It is hoped that readers in this field will be able to understand clearly.
Comments 4: The character v in Equation (1) should be capitalized.
Response 4:The character V in Eq.(1) has been capitalized.Thank you for pointing this out.
Comments 5: The subsection "3.1) Subsection" is not necessary. 
Response 5:The “3.1Section” has been deleted.Thank you for pointing this out.
Comments 6: The authors should comment that they assume that tau_d = tau_f.
Response 6:The definition of the time constant is stated in line 71 to line 72, and the supplementary explanations are newly provided from line 98 to line 99. For more detailed sources, readers need refer to the reference [20].
Comments 7: I cannot catch the meaning of the sentence "Ito can promote spiral wave breakup or stable in AP models." on the 120th line.
Response 7:This sentence does not cite any references, which indeed leads to unclear expression. My revision is to delete this sentence and add a reference at the end of the previous sentence. This reference mainly describes five types of AP models in which Ito can promote spiral wave breakup or stable.
Comments 8: I do not know why the authors chose the parameters in Figure 3: G_si and G_to are both varied and thus the dependency is not clear. Moreover, is the pattern in Figure 3c stable? I found the small structure near the spiral core.
Response 8:Many studies have shown that complex spatiotemporal spiral wave dynamics can be caused by calcium (Ca2+-driven dynamical instabilities, and the stability of spiral waves can be altered by modulating calcium ion conductance[2,7]. Additionally, the transient outward K+current (Ito ) is also a key current that promotes instability of spiral waves in cardiac tissue[29,30]. Therefore, this article focuses on studying the interaction and regulatory mechanisms of these two key currents. There is an error in the citation of the figure, I sincerely apologize for that. The correct statement is “Under the action of Ito , increasing the ISi current can cause it breakup ( Figure 3A and Figure 3C) or promote spiral wave stabilization (Figure 3B and Figure 3D).Thank you to point it out.

Reviewer 4 Report
Comments and Suggestions for Authors
Please see the attachment.

Author Response
We sincerely appreciate the time and effort you have invested in reviewing our work. Your insightful comments have significantly helped us improve the quality of our paper. Below, we provide detailed responses to each of your comments and concerns.
Comments 1: The use of terms like “gate variable” could be briefly defined in the abstract for a more general audience.
Response 1:We have added the definition of the gating variable in both the abstract and the main text, please see line 14 and lines 69 to 72.
Comments 2: The equations could be presented more clearly. For example, add some space around the
equal signs and use proper mathematical typesetting if possible.
Response 2: Equation 1 and Equation 2 have already been standardized in format.
Comments 3: When names and models are mentioned in the article, such as line 64, the Hodgkin-Huxley
model is suggested to list the corresponding references.
Response 3:The Hodgkin-Huxley model is listed the corresponding references[33,34].
Comments4:The English in the manuscript needs improvement again, like
"When is small, a spiral wave...", "current can promote spiral wave breakup when"
Response 4: Do you mean these two sentences? Maybe it's because the formula symbols are not displayed. The original text is as follows. When is small, a spiral wave in 2D tissue is stable with shortwave lengths for smaller calcium ion current leads to smaller repolarization region. current can promote spiral waves breakup when as we can see in Figure 2B, and then transition to stable again as increase.

Round 2
Reviewer 3 Report
Comments and Suggestions for Authors
The authors have revised their manuscript, and it has improved a lot. They have addressed most of my previous comments. However, my concern is on the title of the manuscript. "Ito" is the term in the equation, and it is improper to include the expression in the equation as a part of the title. I think I_{to} is the outward potassium current, and the title consists of the words but not the term in the equation. In addition, "Ito" is misleading since Ito is the family name of a famous Japanese scientist in statistical physics. In the introduction, the authors do not mention I_{to} at all, though "Ito-induced" looks like an essential keyword in the manuscript.
The authors should reconsider the title and introduction part.
Author Response
Comments:
The authors have revised their manuscript, and it has improved a lot. They have addressed most of my previous comments. However, my concern is on the title of the manuscript. "Ito" is the term in the equation, and it is improper to include the expression in the equation as a part of the title. I think I_{to} is the outward potassium current, and the title consists of the words but not the term in the equation. In addition, "Ito" is misleading since Ito is the family name of a famous Japanese scientist in statistical physics. In the introduction, the authors do not mention I_{to} at all, though "Ito-induced" looks like an essential keyword in the manuscript.
The authors should reconsider the title and introduction part.
Response:
Thank you for your valuable feedback regarding our manuscript. We greatly appreciate the time and effort you have invested in reviewing our work. With respect to the title of the manuscript, we completely understand your concern regarding the use of "Ito" in the equation as part of the title, which may cause confusion. You are correct that I_{to} represents the outward potassium current, and including this expression directly in the title might not be appropriate. Additionally, we recognize the potential confusion with the name of the renowned Japanese scientist in statistical physics, Dr. Kiyoshi Itô. To address these issues, we have revised the title to ensure it is composed of clear and descriptive words rather than mathematical terms. The new title is as follows: Impact of Calcium and Potassium Currents on Spiral Wave Dynamics in the LR1 Model.
Regarding the introduction section, we agree that the term "Ito-induced" plays an essential role in the manuscript. However, as you pointed out, we did not adequately introduce or explain I_{to} within the text. We have now revised the introduction to I_{to},which should provide a clearer context for readers and avoid any possible misunderstandings.(line 41-44)
We hope these revisions align with your expectations and enhance the clarity and quality of our manuscript. Please do not hesitate to let us know if further adjustments are necessary.Thank you once again for your insightful comments and guidance.

Round 3
Reviewer 3 Report
Comments and Suggestions for Authors
The authors have adequately addressed my comments in the second stage. The manuscript has improved, and the present version is now suitable for publication in Entropy as it is.